# Analysis of prevalence and influencing factors of stroke in elderly hypertensive patients: Based on the screening plan for the high-risk population of stroke in Jiading District, Shanghai

**Jiefeng Liu[1], Yuqian Chen[2], Chunlin Jin[2], Duo Chen[2], Guangfeng Gao[3], Fen Li[2]***

**1** Department of Epidemiology and Health Statistics, Xiangya School of Public Health, Central South University, Changsha, Hunan, China, **2** Shanghai Health Development Research Center, Jing'an District, Shanghai City, China, **3** Health Information Center of Jiading District, Shanghai City, China

* lifen@shdrc.org

## Abstract

### Background

The purpose of this study is to investigate and analyze the prevalence and influencing factors of stroke in hypertensive patients aged 60 and above in Jiading District, Shanghai.

### Methods

The population-based study included 18,724 screened people with hypertension (age $\geq$ 60 years, 48.7% women). From 2016 to 2019, data on demographics, potential influencing factors and health status were collected through face-to-face interviews, physical examinations, and laboratory tests. Logistic multivariate logistic regression model was used to analyze the influencing factors associated with stroke.

### Results

Among the object of study from 2016 to 2019, 2,025 patients were screened for stroke, with the overall prevalence rate of 10.82% (10.41%-11.23%). Multivariate adjusted model analysis showed that dyslipidemia (OR:1.31,95%CI:1.19–1.45), lack of exercise (OR:1.91,95% CI:1.32–2.76), atrial fibrillation [OR:1.49,95%CI:1.35–1.65], family history of stroke (OR:2.18,95%CI:1.6–2.88) were the significant independent influencing factors of stroke in hypertensive patients over 60 years old. When these four factors were combined, compared with participants without any of these factors, the multi-adjusted odds ratios (95% confidence interval) of risk of stroke for persons concurrently having one, two and three or more of these factors were 1.89 (1.67–2.13), 2.15 (1.86–2.47) and 6.84 (4.90–9.55), respectively (linear trend P < 0.001); after multivariate adjustment, the family history of stroke had additive interaction with lack of exercise [RERI = 1.08(0.22–1.94), AP = 0.19(0.04–0.35),

**Data Availability Statement:** All relevant data are within the manuscript and its Supporting Information files.

**Funding:** This work was supported by Outstanding Academic Leader Program and the Outstanding Young Medical Personnel Training Program of the Shanghai Municipal Health Commission (grant no. 2018YQ51,to Fen Li) and National Natural Science Funds of China'Empirical Study on Evaluation 430 of Integrated Stroke Prevention and Treatment System Based on Real World Data' (Grant No. 43172004138).

**Competing interests:** The authors have declared that no competing interests exist.

S = 1.31(1.02–1.69)], dyslipidemia [RERI = 0.87(0.41–1.33), AP = 0.23(0.08–0.38), S = 1.46(1.04–2.05)].

## Conclusion

The prevalence of stroke was high in hypertensive patients aged 60 and above in Jiading District, Shanghai. Dyslipidemia, lack of exercise, atrial fibrillation and family history of stroke were significantly associated with stroke in hypertensive population. Stroke risk can be increased especially when multiple factors coexisting, and family history of stroke combined with a lack of exercise or dyslipidemia.

## Introduction

Stroke is a complex disease affected by many factors, some which may be independent, but also may interact with each other. The pathogenesis and risk factors of stroke in different regions and different populations are not completely clear, and the influencing factors of stroke in people with different characteristics (gender, age, and underlying diseases) are not completely consistent [1, 2]. Stroke is one of the main causes of morbidity and mortality in people with hypertension. Hypertension is the most important and changeable risk factor of stroke. Consequently, it is important to investigate the factors that influence stroke in hypertensive patients and to implement targeted intervention strategies [3]. However, in different studies, the influencing factors of stroke in patients with hypertension are inconsistent, which may be related to region and race. Moreover, Few studies have examined at the interaction between multiple risk factors and stroke, which is important because risk factors often coexist in middle-aged and elderly people [4]. Therefore, this study has examined the prevalence of stroke, influencing factors and their relationship in hypertensive patients, based on a large-scale and high-quality screening program for high-risk population of stroke in Jiading District of Shanghai, in order to provide more evidence for precise intervention required to reduce the risk of stroke.

## Materials and methods

### Research design

The research design of this survey is based on a screening and intervention plan for stroke high-risk populations in a certain district of Shanghai. The plan was implemented in 2016. It provides screening services for people at high risk of stroke for residents over 35 years of age who are under the health management of chronic diseases in the community, and implements full coverage screening for key populations (populations with hypertension, diabetes, and heart diseases). The screening program is implemented in two phases. The first phase was carried out in the local community center. Residents aged 35 and up were interviewed by qualified medical professionals using a self-made standard questionnaire. The standard questionnaire included demographic, medical history, family history of stroke, medication history, lifestyle, and vascular influencing factors. Educational background, marital status, occupation was also recorded. Weight, height, waist circumference and blood pressure were measured and recorded according to standard procedures. Participants with a history of stroke were examined at the scene by a professional neurologist for confirmation. The second stage was to collect data that cannot be obtained in the field, including ECG (electro cardio gram),

fasting blood glucose and serum lipid groups, such as low-density lipoprotein cholesterol (LDL-C), high-density lipoprotein cholesterol (HDL-C), triglycerides and total cholesterol.

## Study population

Since 2016, a comprehensive screening of all chronic patients has been conducted in the Chronic Disease Management Bank of Jiading District, Shanghai, as well as a community-wide survey. The plan is mainly based on the district's chronic disease registration management database. It registers all diagnosed chronic disease patients in the district. According to the principle of "informed, consent, and voluntary", the health department and the community strengthen publicity and guidance. All hypertension patients over 35 years old who met the required criteria were completely screened. The required criteria as follows: 1) permanent residents over 35 years old who had lived in the area for at least 6 months; 2) diagnosed as hypertensive patients; 3) no contraindications to screening; and 4) able to sign informed consent with their guardians and voluntarily participate in all screening programs. And as of 2019, 26,174 hypertensive patients were randomly selected as survey participants, and after excluding people who are unwilling to cooperate and who did not meet the required criteria, we were able to screen 21,902 hypertensive patients, with a screening response rate of 83.69%. We recorded the potential influencing factors of stroke and the occurrence of stroke in these screened participants. In the end, we settled on a sample population of 18,724 people aged 60 and up who had hypertension.

## Assessment and definition of variables

1. Hypertension: any of the following criteria can be met: (a) History of hypertension (Provided a hospital diagnosis certificate). (b) The results of this screening showed that blood pressure was increased (systolic blood pressure 140 mmHg or diastolic blood pressure 90 mmHg).

2. Atrial fibrillation or valvular heart disease: anyone of the following can be met: (a) Previous medical history (Provided a hospital diagnosis certificate). (b) The patient's electrocardiogram (ECG) in this screening showed atrial fibrillation. The diagnostic basis must meet the following criteria: regular and orderly atrial electrical activity has disappeared, replaced by rapid and disordered fibrillation waves on the electrocardiogram. ECG showed that P wave disappeared and replaced by F wave. The R-R interval was 5.

3. Smoking: People who smoke continuously or cumulatively for 6 months or more in a lifetime are defined as smokers.

4. Dyslipidemia: any of the following criteria can be met: (a)Previous medical history (Provided a hospital diagnosis certificate). (b) In this field survey, total cholesterol was 6.22 mmol/L (240 mg/D), and glyceride was more than 2.26 mmol/L. (200 mg/DD), high density lipoprotein < 1.04 mmol/L (40 mg/D), one or more of the abnormalities can determine dyslipidemia.

5. Diabetes: any of the following criteria can be met: (a) Patients with previous medical history (Provided a hospital diagnosis certificate) and receiving diabetes drugs or insulin treatment. (b) The field measurement showed high blood glucose (random blood glucose ≥11.0mmol/l or fasting blood glucose ≥7.0mmol/l).

6. Lack of exercise: The standard refers to CHNS Reference [5], exercise more than 3 times a week, moderate intensity or more than 30 minutes each time or engaged in moderate and

severe physical workers are regarded as regular physical exercise; otherwise, it is lack of exercise.

7. Family history of stroke: Provided a hospital diagnosis certificate, The diagnosis is based on anyone in the immediate family (grandfather, grandmother, parents and siblings) who has ever been diagnosed with a stroke [6].

8. obesity: BMI $\geq$28 kg/m$^2$ and overweight: BMI of 24–27.9 kg/m$^2$.

9. Stroke: The diagnosis of stroke is mainly based on the diagnostic criteria of the WHO collaborative group [7], combined with the surveyed person's reports on previous diagnoses, and the patient is required to provide corresponding medical documents or imaging data.

10. Course of hypertension (years): The amount of years between a patient's diagnosis of hypertension and the screening.(11)Education: (a) Primary school or illiterate refers to having only received the most basic elementary education of 6 years or less in China; (b) have completed a 9-year compulsory education or higher education is referred to as secondary school and above in China.

## Data statistics

A cross-sectional study was conducted in the hypertensive patients who received the high-risk screening for stroke. We used IBM SPSS Statistics V22.0 for Windows (IBM Corp. Released 2013, Armonk, NY: IBM Corp) for all analyses. All characteristics between participants with and without stroke were compared using t-test for continuous variables and $\chi^2$ test for categorical variables. Multivariable logistic model was used to estimate the odds ratios (OR) and 95% confidence intervals (CI) of stroke associated with individual influencing factors and their load, which was assessed by counting the number of influencing factors that were significantly related to an increased odds ratio of stroke (P$\leq$0.05). We reported the results from two models: Model 1 was adjusted for age and gender, while Model 2 was adjusted for education, and if applicable, for course of hypertension, diabetes, dyslipidemia, smoking status, use of antihypertensive drugs, overweight or obesity, few physical exercises, marriage status and Atrial fibrillation or valvular heart disease.

To evaluate the additive interaction between two factors, we used relative excess risk due to the interaction (RERI), which was also referred to as the interaction contrast ratio (ICR) without exposure and attribute proportion due to an interaction (AP) with both exposures. The specific analysis methods are as follows: RERI = $RR_{A1B1}$-$RR_{A1B0}$-$RR_{A0B1}$+1; ②AP = RERI/$RR_{A1B1}$; ③S = ($RR_{A1B1}$-1)/[($RR_{A0B1}$-1)+ ($RR_{A0B0}$-1)], A1 and A0 indicate the exposure and non-exposure of this factor respectively, while B1 and B0 are the same. If there is no additive interaction between the two variables, the RERI and AP confidence intervals should all be 0, while the S confidence interval should be 1. The calculation method of interaction is based on the construction and algorithm of interaction analysis index proposed by Rothman [8, 9], Hosmer and Lemeshow [10], the evaluation index of interaction is further calculated by logistic regression model, and the confidence interval of additive interaction is estimated by introducing excel calculation table compiled by Andersson [11]. Detailed information on an additive interaction has been published elsewhere [11–13]. Statistical significance was defined as two-tailed P<0.05.

## Ethics approval and consent to participate

The study was complied with the Declaration of Helsinki and was approved by the institute review board of Shanghai Health Development Research Center. All study participants provided written

informed consent before enrolling in our study. Moreover, the informed consent was obtained from parents/legal representative (LARs) for participants with no education.

## Results

### Characteristics

The average age of all study participants was 69 years old (mean: 69, standard deviation: 9.1), with 48.7% of them were male. Compared with people without stroke, the stroke patients were older, had lower education, higher marriage rate and longer duration of hypertension, as well as being more likely to have atrial fibrillation, dyslipidemia, family history of stroke and lack of exercise (P < 0.001). Conversely, it showed no differences in smoking, BMI, overweight or obesity, and diabetes (Table 1).

### Prevalence of stroke

A total of 18,724 hypertensive patients were screened, with 2,025 of them having a stroke. The prevalence of stroke among people aged 60 and over was 10.8% (10.41%-11.23%). The prevalence of stroke was 7.0% in those aged 60–69, 12.7% for those aged 70–79, and 17.9% for those aged 80 and up, respectively. There is no gender difference in prevalence.

### Influencing factors of stroke

Multivariable logistic regression analysis showed that dyslipidemia (OR:1.31,95%CI:1.19–1.45), lack of exercise (OR:1.91,95%CI:1.32–2.76), atrial fibrillation [OR:1.49,95%CI:1.35–1.65], family

**Table 1. Demographic information and clinical characteristics of study population.**

| Characteristics | Population | Stroke | | |
|---|---|---|---|---|
| | n = 18724 | no (n = 16699) | yes (n = 2025) | P* |
| Age (years), mean±SD | 68.98 ±9.12 | 68.51±9.12 | 73.35±7.86 | <0.001 |
| Sex, n (%) | 9118(48.70) | 8139(48.74) | 979(48.34) | 0.541 |
| Marriage status, n (%) | | | | |
| Married | 17551(93.74) | 15695(93.99) | 1856(91.7) | <0.001 |
| Unmarried / divorced / widowed | 1173(6.26) | 1004(6.01) | 169(8.35) | |
| Education, n (%) | | | | |
| Primary school or illiterate | 9855(52.63) | 8639(51.73) | 1216(60.05) | <0.001 |
| Secondary school and above | 8869(47.37) | 8060(48.27) | 809(39.95) | |
| Course of hypertension (years) | | | | |
| 0–5 | 9323(49.80) | 8392(50.25) | 931(46.0) | <0.001 |
| 6–10 | 6497(34.70) | 5773(34.57) | 724(35.8) | |
| >10 | 2904(15.51) | 2534(15.17) | 370(18.3) | |
| BMI (kg/m2), mean±SD | 24.93±4.61 | 24.92±4.71 | 24.72±3.53 | 0.071 |
| Smoking, n (%) | | | | |
| yes | 5677(30.32) | 5073(30.38) | 604(29.83) | 0.414 |
| no | 13039(69.64) | 11626(69.62) | 1421(70.17) | |
| Taking anti-hypertensive drugs, n (%) | 17893(95.56) | 15953(95.53) | 1940(95.80) | 0.604 |
| Atrial fibrillation, n (%) | 288(1.54) | 218(1.31) | 70(3.46) | <0.001 |
| Overweight or obese, n (%) | 6200(33.11) | 5553(33.25) | 647(31.95) | 0.115 |
| Dyslipidemia, n (%) | 5798(31.96) | 5084(30.44) | 714(35.26) | <0.001 |
| Lack of exercise, n (%) | 10811(57.74) | 9435(56.50) | 1376(67.95) | <0.001 |
| Diabetes, n (%) | 7066(37.74) | 6328(37.89) | 738(36.44) | 0.207 |
| Family history of stroke, n (%) | 391(2.08) | 293(1.75) | 98(4.84) | <0.001 |

**Table 2. Multivariable logistic regression analysis for influencing factors of stroke.**

| Characteristics | Total population | Number of stroke patients | Odds Ratio (95% confidence interval) | |
|---|---|---|---|---|
| | | | Model 1 | Model 2 |
| Course of hypertension (years) | | | | |
| 0–5 | 9323 | 931 | 1.00 (Reference) | 1.00 (Reference) |
| 6–10 | 6497 | 724 | 1.09 (0.98–1.21) | 1.08 (0.97–1.20) |
| >10 | 2904 | 370 | 1.00 (0.87–1.15) | 1.00 (0.88–1.16) |
| Smoking | | | | |
| no | 13047 | 1421 | 1.00 (Reference) | 1.00 (Reference) |
| yes | 5677 | 604 | 1.16 (1.01–1.32)* | 1.12 (0.97–1.29) |
| Marriage status | | | | |
| Married | 17551 | 1856 | 1.00 (Reference) | 1.00 (Reference) |
| Unmarried / divorced / widowed | 1173 | 169 | 1.00 (0.84–1.19) | 0.95 (0.82–1.10) |
| Taking antihypertensive drugs | | | | |
| No | 831 | 85 | 1.00 (Reference) | 1.00 (Reference) |
| Yes | 17893 | 1940 | 0.92 (0.73–1.16) | 0.94 (0.75–1.18) |
| Atrial fibrillation | | | | |
| No | 18436 | 1955 | 1.00 (Reference) | 1.00 (Reference) |
| Yes | 288 | 70 | 2.36 (1.79–3.12)** | 2.18 (1.6–2.88)** |
| Diabetes | | | | |
| No | 11658 | 1287 | 1.00 (Reference) | 1.00 (Reference) |
| Yes | 7066 | 738 | 0.98 (0.89–1.08) | 0.99 (0.90–1.09) |
| Overweight or obese | | | | |
| No | 12524 | 1378 | 1.00 (Reference) | 1.00 (Reference) |
| Yes | 6200 | 647 | 1.02(0.93–1.13) | 1.00 (0.90–1.10) |
| Dyslipidemia | | | | |
| No | 12926 | 1311 | 1.00 (Reference) | 1.00 (Reference) |
| Yes | 5798 | 714 | 1.35 (1.23–1.49)** | 1.31 (1.19–1.45)** |
| Lack of exercise | | | | |
| No | 7913 | 649 | 1.00 (Reference) | 1.00 (Reference) |
| Yes | 10811 | 1376 | 1.55(1.41–1.71)** | 1.49 (1.35–1.65)** |
| Family history of stroke | | | | |
| No | 18333 | 1927 | 1.00 (Reference) | 1.00(Reference) |
| Yes | 391 | 98 | 3.44 (2.71–4.36)** | 3.47 (2.76–4.37)** |

Model 1: adjusted for age and gender; Model 2 adjusted for age, gender, education level, duration of hypertension, antihypertensive drugs, smoking, lack of exercise, dyslipidemia, diabetes, dyslipidemia, overweight or obesity, atrial fibrillation, and family history of stroke

*P<0.05

**P<0.001.

history of stroke (OR:2.18,95%CI:1.6–2.88) were significantly related to prevalence of stroke in hypertensive population over 60 years old, even after adjusting for multiple factors (model 2), the results were still stable (Table 2).

When analyzing the relationship between the number of factors and stroke, the results showed that compared with the population without these factors, the multi factor adjusted odds ratio (OR) and 95% confidence interval of stroke was 1.89 (1.67–2.13), 2.15 (1.86–2.47) and 6.84 (4.90–9.55), respectively. There was a linear trend relationship between the risk and the number of coexisting influencing factors (linear trend P < 0.001) (Table 3).

**Table 3. The analysis result of the relationship between the number of influencing factors and the prevalence of stroke.**

| Number of influencing factors | Total number of subjects | Number of stroke patients | Odds Ratio (95% confidence interval) | |
|---|---|---|---|---|
| | | | Model 1 | Model 2 |
| continuous variable (0–4) | 18723 | 2025 | 1.50 (1.41–1.60)** | 1.49 (1.40–1.59)** |
| Categorical variable | | | | |
| 0 | 5662 | 369 | 1.00 (reference) | 1.00 (reference) |
| 1 | 9017 | 1110 | 1.91 (1.69–2.16) | 1.89 (1.67–2.13) |
| 2 | 3864 | 492 | 2.21 (1.92–2.54)* | 2.15 (1.86–2.47)* |
| ≥3 | 180 | 54 | 7.02 (5.05–9.76)** | 6.84 (4.90–9.55)** |
| Linear trend test | | | <0.001 | <0.001 |

Model 1: adjusted for age and gender; Model 2: adjusted for age, gender, education level, duration of hypertension, antihypertensive drugs, smoking, lack of exercise, dyslipidemia, diabetes, dyslipidemia, overweight or obesity, atrial fibrillation, and family history of stroke

*P<0.05

**P<0.001.

## Interaction of influencing factors

After adjusting for multiple factors, family history of stroke was found to have an additive interaction with lack of exercise, and dyslipidemia (Table 4). The result of additive interaction analysis showed that the prevalence of stroke caused by the interaction between family history of stroke and lack of exercise is 1.08 times higher than other unknown factors and contributed to 19% of the overall effect with interaction index (SI) 1.31 (1.02–1.69). Likewise, the prevalence of stroke caused by a combination of dyslipidemia and a family history of stroke is 0.87 times higher than that caused by unknown causes. The interaction contributed to 23% of the overall effects of these two variables, and the interaction index (SI) is 1.46. (1.04–2.05).

## Discussion

In this population-based study, we found that the prevalence and OR (95% confidence interval) of stroke in the elderly with hypertension in Jiading District, Shanghai was 10.82%

**Table 4. Analysis of additive interaction of influencing factors of stroke.**

| Factor 1 | Factor 2 | AOR※(95% CI) | RERI | AP | SI |
|---|---|---|---|---|---|
| Lack of exercise | Family history of stroke | | 1.08 | 0.19 | 1.31 |
| | | | (0.22–1.94) | (0.04–0.35) | (1.02–1.69) |
| No | No | 1.00 | | | |
| Yes | No | 1.80(1.11–2.92) | | | |
| No | Yes | 3.68(2.68–5.04) | | | |
| Yes | Yes | 5.55(4.02–7.67) | | | |
| Dyslipidemia | Family history of stroke | | 0.87 | 0.23 | 1.46 |
| | | | (0.41–1.33) | (0.08–0.38) | (1.04–2.05) |
| No | No | 1.00 | | | |
| Yes | No | 1.01(0.63–1.62) | | | |
| No | Yes | 2.88(1.99–4.18) | | | |
| Yes | Yes | 3.76(2.61–5.42) | | | |

AOR: adjusted odds ratio; RERI: the relative risk due to interaction; AP: the attributable proportion; S: the synergy index

※Adjusted for age, gender, education, marital status, smoking, overweight or obese, taking antihypertensive drugs, duration of hypertension.

(10.41%-11.23%). The prevalence of stroke was significantly associated with dyslipidemia, atrial fibrillation, lack of exercise and family history of stroke. The linear trend test showed the increase in the number of coexisting influence factors with the high risk of stroke. We discovered that there could be an additive association between family history of stroke and lack of exercise, dyslipidemia, and atrial fibrillation when we investigated at the interaction between the four significant influencing factors mentioned above. We found that the prevalence of stroke in the elderly with hypertension in Jiading District of Shanghai was 10.8%, which was remarkably similar to the findings of another rural survey in China [14]. This result was about 2–3 times higher than the standardized prevalence of stroke in general population reported by other Chinese studies [15, 16], and much higher than the stroke prevalence of people with diabetes and people with atrial fibrillation reported in other Asian studies [17, 18].

Identifying the influencing factors of stroke that can be changed in people with hypertension can help prevent the occurrence of stroke. Our study found that dyslipidemia, atrial fibrillation, lack of exercise, and family history of stroke were independent influencing factors for stroke in hypertensive population. Lack of exercise and dyslipidemia have also been shown to be significantly related to the prevalence of stroke in hypertensive patients in some studies, which is consistent with our findings [19, 20]. In addition, several studies have reported that alcohol consumption, excessive salt intake, uncontrolled blood pressure, high carotid artery diameter and plaque may all be risk factors for stroke in hypertensive population [19–22]. Whereas, another study reported that high fasting blood glucose is an risk factor for stroke in hypertensive population [19], which contradicts our findings. The contradictory results may be due to different locations of investigation. The blood glucose control of diabetic patients in this investigation is relatively good, so we did not find that diabetes are significantly related to stroke. Dyslipidemia, atrial fibrillation, and lack of exercise were also risk factors for stroke in people with type 2 diabetes, according to a cohort study in Korea [23], suggesting that these three factors could be common risk factors for stroke in people with hypertension and people with diabetes.

We found that the number of coexisting influencing factors in hypertensive population is directly proportional with the risk of stroke, which is appropriate with other studies [24]. This study also reported that the increase of stroke risk factors is independently associated with the increase of long-term mortality. Some large prospective cohort studies have found that the risk of cardiovascular and cerebrovascular disease increased with the number of coexisting risk factors [25, 26]. In some regions, people with three or more influencing factors are classified as high-risk groups, and targeted interventions are implemented. A latest study shows that there is a linear association between the number of cardiovascular influencing factors and the risk of asymptomatic intracranial atherosclerotic stenosis, and asymptomatic intracranial atherosclerotic stenosis is a major cause of stroke, which may be one of the explanations for the linear relationship between the number of stroke influencing factors and the occurrence of stroke [27]. Moreover, our study found that there was additive interaction between family history of stroke and lack of exercise, dyslipidemia, and atrial fibrillation in hypertensive population indicating there may be a biological relationship between family history of stroke and these three factors. There are currently few studies examining the interaction between the important influencing factors of stroke, so we believe that further research is required to better understand the interaction between the influencing factors of stroke and their mechanisms. More importantly, it suggests that the primary prevention of stroke in hypertensive population should focus on those with family history of stroke accompanied by lack of exercise, dyslipidemia, and atrial fibrillation.

There are several limitations to this study. Firstly, since this is a cross-sectional study, the causal relationship between these factors and stroke cannot be determined. Second, variables

such as alcohol consumption, dietary patterns, blood pressure control, and carotid artery condition were not included in our research, which are all important factors affecting stroke and may have influenced our results. Finally, we were unable to differentiate between ischemic and hemorrhagic strokes due to our inability to accurately determine the form of stroke, which may have impacted our findings. Therefore, we also expect to have large, high-quality prospective cohort studies to explore and verify the influencing factors of stroke in hypertensive population.

## Conclusion

The prevalence of stroke was high in hypertensive population aged 60 and above. Dyslipidemia, lack of exercise, atrial fibrillation and family history of stroke were significantly associated with stroke in hypertensive population, especially when multiple influence factors coexisting, and family history of stroke coexisting with other significant risk factors. These results show that the risk of stroke in different groups is quite varied. The primary intervention of stroke should focus on those who have multiple influencing factors or the interaction of accompanying factors, which may more effectively prevent or delay the process of stroke in hypertensive population.

## Supporting information

**S1 Dataset.**
(XLS)

## Author Contributions

**Conceptualization:** Fen Li.

**Data curation:** Jiefeng Liu.

**Formal analysis:** Jiefeng Liu, Yuqian Chen, Fen Li.

**Investigation:** Yuqian Chen, Duo Chen, Guangfeng Gao.

**Methodology:** Jiefeng Liu, Chunlin Jin, Fen Li.

**Resources:** Chunlin Jin, Guangfeng Gao.

**Software:** Guangfeng Gao.

**Supervision:** Duo Chen.

**Validation:** Duo Chen.

**Writing – original draft:** Jiefeng Liu, Fen Li.

**Writing – review & editing:** Jiefeng Liu, Yuqian Chen, Chunlin Jin, Fen Li.

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
