## [Decision Letter · Decision Letter 0]

5 Feb 2021

PONE-D-20-40953

Analysis of prevalence and risk factors of stroke in elderly hypertensive patients Based on the screening plan for the high-risk population of stroke in Jiading District, Shanghai

PLOS ONE

Dear Dr. Li,

Thank you for submitting your manuscript to PLOS ONE. After careful consideration, we feel that it has merit but does not fully meet PLOS ONE’s publication criteria as it currently stands. Therefore, we invite you to submit a revised version of the manuscript that addresses the points raised during the review process.

We look forward to receiving your revised manuscript.

Kind regards,

Yiqiang Zhan

Academic Editor

PLOS ONE

Journal Requirements:

- https://www.nature.com/articles/s41598-017-06691-1

- https://onlinelibrary.wiley.com/doi/10.1111/ene.14144

- https://pesquisa.bvsalud.org/gim/?lang=en%2Cau%3A%22Martins+Neto%2C+Viviana%22&q=au%3A%22Jiaqi+XU%22

In your revision ensure you cite all your sources (including your own works), and quote or rephrase any duplicated text outside the methods section. Further consideration is dependent on these concerns being addressed.

Reviewers' comments:

Reviewer's Responses to Questions

**Comments to the Author**

1. Is the manuscript technically sound, and do the data support the conclusions?

Reviewer #1: Partly

Reviewer #2: No

2. Has the statistical analysis been performed appropriately and rigorously? 

Reviewer #1: Yes

Reviewer #2: No

3. Have the authors made all data underlying the findings in their manuscript fully available?

Reviewer #1: Yes

Reviewer #2: No

4. Is the manuscript presented in an intelligible fashion and written in standard English?

Reviewer #1: No

Reviewer #2: No

5. Review Comments to the Author

Reviewer #1: In this population-based cross-sectional study, the authors characterized the prevalence of stroke among hypertensive patients aged 60+ and explored the relationships between stroke and a wide range of demographic, lifestyle and clinical variables. The main strengths include the sample size of study population, richness of studied risk factors, and the statistical analysis where potential interaction among stroke-associated variables was tested. However, there are also limitations in study design, analysis and result discussion, as specified below.

1. Introduction: motivation of the study was not clearly explained.

2. Materials and Methods: A subtitle "Study design" should be included first thing first.

2.1 Study population: I suggest the authors to include a flowchart and explicitly explain the changes of Ns in each step. For instance, what does it mean by "26174 hypertension (HT) patients have been filed in the area" (line 77), it is astonishing to see only 26k hypertensive patients in Jiading if the analyzed database covered "all the diagnosed chronic disease patients in the area". Why only 21,902/26,174 patients were screened for stroke? Is age >=60 the only inclusion criteria when selecting 18,724 HT patients for final analysis?

2.2 Definitions: a) The authors might wish to separately define the outcome variable and the explanatory variables. b) Please avoid being ambiguous or non-informative in definition (e.g. "The ECG showed atrial fibrillation" in line 95). c) What is the rationale for defining 'lack of exercise' as it was presented? d) Definition for education and its rationale? And most importantly, e) I assume all data were collected cross-sectionally, but 'risk factors' (implying temporality) were used throughout the text. Please clarify.

2.3 Statistics: in line 121, what is 'Baseline'? Line 122, it should be 'multivariable logistic models'. Line 124, it is unclear what 'load' is and how it was assessed. In addition, the section for interaction calculation is unclear.

3. Results

3.1 Characteristics: A minor suggestion to use consistent term, say 'study population', when referring to the analyzed study participants. Throughout the text, particularly in Tables, please uniform the rounding rules and reporting of p-values (sometimes full results, sometimes above/below .05 etc.). Again, in title of Table 1, what is 'Baseline'?

3.2 Prevalence of stroke: Please revise "There was no gender difference between men and women". It is also confusing to use 'different population' in title for Table 2, consider population strata or alike.

3.3 Risk factors of stroke: a) Why not use BMI instead of binary variable obesity? b) Why smoking was adjusted as binary variable while it was defined as active smoker and quitter? c) Why education was not adjusted in either model? d) Was there correlation among included explanatory variables, i.e. between BMI, exercise and dyslipidemia? If yes, were they taken into consideration in model fitting and in counting number of 'risk factors'? I wonder results in Table 4 may change if dimentionality of included variables could have been reduced.

3.4 Interaction of risk factors: I suggest the authors to explain a bit more about Table 5, such as by comparing across PERI, AP and SI (not 'S').

4. Discussion and Conclusion

Overall, I appreciate the in-depth discussion of the main findings, but please shorten the discussion substantially. Besides, any conclusion about risk of stroke should be avoid given the cross-sectional design of the study. Therefore, I would prefer not to use 'risk factors' to denote the analyzed explanatory variables throughout the article either.

The manuscript should also undergo a thorough language check to minimize grammatical errors and/or typo and to further improve readability.

Reviewer #2: This manuscript aimed to describe the prevalence of stroke in elderly population with hypertension in Shanghai and explore the risk factors of stroke. With a cross-sectional design, the authors employed multivariate logistic regression model to analyze the risk factors of stroke among hypertension population.

The results section, page 12 line 141 for those without stroke, 94.0% were married, while for those with stroke 91.7% were married. But the author said stroke patients had higher marriage rate.

This manuscript is poor written. These are many typos and misuse of words. For instance,

page9 line 50, “among which are independent, but also interact with each other”,

page 10 line 87, what is ECG? Should give a full name when it occurred first time.

page 11 line 111 “immediate family”?

page 13 table 1 “secondary school?”

The titles of all the tables need to be edited, it should clear describe what will be presented in the table.

6. PLOS authors have the option to publish the peer review history of their article (what does this mean?). If published, this will include your full peer review and any attached files.

Reviewer #1: No

Reviewer #2: No

---

## [Author Response · Author response to Decision Letter 0]

14 Jun 2021

Reviewer #1: In this population-based cross-sectional study, the authors characterized the prevalence of stroke among hypertensive patients aged 60+ and explored the relationships between stroke and a wide range of demographic, lifestyle and clinical variables. The main strengths include the sample size of study population, richness of studied risk factors, and the statistical analysis where potential interaction among stroke-associated variables was tested. However, there are also limitations in study design, analysis and result discussion, as specified below.

1. Introduction: motivation of the study was not clearly explained.

Response: Thank you very much for your suggestion. The detailed revisions can be found in Introduction. 

2. Materials and Methods: A subtitle "Study design" should be included first thing first. 

Response: Thank you very much for your suggestion. The subtitle "Study design" has been added in first thing first.

2.1 Study population: I suggest the authors to include a flowchart and explicitly explain the changes of Ns in each step. For instance, what does it mean by "26174 hypertension (HT) patients have been filed in the area" (line 77), it is astonishing to see only 26k hypertensive patients in Jiading if the analyzed database covered "all the diagnosed chronic disease patients in the area". Why only 21,902/26,174 patients were screened for stroke? Is age >=60 the only inclusion criteria when selecting 18,724 HT patients for final analysis?

Response: Thanks for your constructive comments. According to your questions, we described the "study population" section more accurately and detailed how the research population was obtained. The specific description is as follows. Since 2016, a comprehensive screening of all chronic patients in the Chronic Disease Management Bank of Jiading District, Shanghai has been conducted, and the general survey was performed on a community basis. The plan is mainly based on the chronic disease registration management database of the district. It registers all diagnosed chronic disease patients in the district. According to the principle of "informed, consent, and voluntary", the health department and the community strengthen publicity and guidance and implement full coverage screening for people aged over 35, meeting the inclusion criteria, and with hypertension registered in the chronic disease management database. The inclusion criteria of survey objects include: 1) permanent residents whose annual residence time is ≥6 months, and are over 35 years old; 2) clearly diagnosed as hypertension patients; 3) no contraindications related to screening; 4) patients who can sign an informed consent form by themselves or accompanied by their guardians and participate in all screening projects voluntarily. As of 2019, 26174 hypertension patients were randomly selected as survey subjects. After excluding people who are unwilling to cooperate and do not meet the inclusion criteria, we finally screened 21,902 hypertension patients, with a screening response rate of 83.69%. For these screened populations, the potential influencing factors of stroke and the occurrence of stroke were recorded. In the end, a total of 18,724 people with hypertension 60 years and older were selected as our study population. At present, the screening project is still ongoing. According to statistics, there are 155,829 hypertensive patients in the chronic disease registration management database in the district. A lot of manpower, material, and financial resources are required to achieve a comprehensive screening. Therefore, valuable information could be obtained through the investigation and analysis of the first stage to guide the follow-up screening and crowd intervention.

Besides, to include enough research populations to ensure that the results are closer to the real world, we did not adopt strict inclusion criteria, and the inclusion criteria and procedures are relatively simple and clear. Thus, flowcharts were not employed to represent our inclusion procedures. Finally, thank you much for your valuable suggestions.

2.2 Definitions: a) The authors might wish to separately define the outcome variable and the explanatory variables. b) Please avoid being ambiguous or non-informative in definition (e.g. "The ECG showed atrial fibrillation" in line 95). c) What is the rationale for defining 'lack of exercise' as it was presented? d) Definition for education and its rationale? And most importantly, e) I assume all data were collected cross-sectionally, but 'risk factors' (implying temporality) were used throughout the text. Please clarify.

Response:ThanksResponse: Thanks for your constructive comments. According to your comments, we have reviewed and revised the definitions of all research variables. For details, please see the "Definitions" section of the article. Additionally, we have also replaced all risk factors in the article with influencing factors.

2.3 Statistics: in line 121, what is 'Baseline'? Line 122, it should be 'multivariable logistic models'. Line 124, it is unclear what 'load' is and how it was assessed. In addition, the section for interaction calculation is unclear.

Response: a) Thanks for your constructive comments. Baseline characteristics refer to All characteristics, including demographic information and potential influencing factors. We apologize for using this term incorrectly, and we have corrected it.

b) Load refers to the burden of multiple factors, which was assessed by counting the number of influencing factors significantly related to an increased odds ratio of stroke (P≤0.05). The ‘load’ has the same usage in other published articles.

c) Besides, we have detailed the calculation of interaction in detail in the article. The specific method is described as follows. To evaluate the additive interaction between two factors, we used relative excess risk caused by the interaction (RERI), which was also called the interaction contrast ratio (ICR) without exposure and attributable proportion due to interaction (AP) with both exposures. The specific analysis methods are: RERI=RRA1Bl-RRA1B0-RRA0B1+1; AP=RERI／RRAlBl; S=(RRAlBl-1)／[(RRA0B1-1)+ (RRA0B0-1)]. A1 and A0 represent exposure and non-exposure of the factor, respectively, and B1 and B0 are the same. If there is no additive interaction between the two factors, the credible interval of RERI and AP should contain 0, and the credible interval of S should contain 1. The calculation method of the interaction is based on the construction and algorithm of the interaction analysis index proposed by Rothman8-9, Hosmer, and Lemeshow10 to further calculate the evaluation index of the interaction through the logistic regression model and introduce the Excel calculation table compiled by Andersson et al. 11 to estimate the confidence interval of the additive interaction. Detailed information on an additive interaction has been published 11,12,13. Statistical significance was defined as two-tailed P<0.05.

3. Results

3.1 Characteristics: A minor suggestion to use consistent term, say 'study population', when referring to the analyzed study participants. Throughout the text, particularly in Tables, please uniform the rounding rules and reporting of p-values (sometimes full results, sometimes above/below .05 etc.). Again, in title of Table 1, what is 'Baseline'?

Response: Thank you very much for your suggestion. We have used uniform terminology in the text to indicate "study population", and P values in all tables have also been modified. Baseline characteristics refer to all characteristics, including demographic information and potential influencing factors. We are sorry for misusing this term, and this error has been corrected.

3.2 Prevalence of stroke: Please revise "There was no gender difference between men and women". It is also confusing to use 'different population' in title for Table 2, consider population strata or alike.

Response: We are sorry for our carelessness. We have changed this sentence to "There is no gender difference in prevalence”. Considering your suggestions and the results of our discussion, we agree that it is not necessary to describe the prevalence of different populations in this section, because the demographic information of stroke and non-stroke populations has been compared in Table 1. In the end, this part of the content was deleted, and only the prevalence descriptions of people of different ages and genders were kept.

3.3 Risk factors of stroke: a) Why not use BMI instead of binary variable obesity? b) Why smoking was adjusted as binary variable while it was defined as active smoker and quitter? c) Why education was not adjusted in either model? d) Was there correlation among included explanatory variables, i.e. between BMI, exercise and dyslipidemia? If yes, were they taken into consideration in model fitting and in counting number of 'risk factors'? I wonder results in Table 4 may change if dimentionality of included variables could have been reduced.

Response: a) We believe that converting BMI into a binary variable of obesity is more conducive to the interpretation of results and the guidance of intervention. We also substituted BMI as a continuous variable into the analysis, and only obtain the same negative result as the obesity variable. 

b): We are sorry for misdescribing the definition of smoking. We have redefined this variable in the text: Study population who smoke continuously or cumulatively for 6 months or more in a lifetime are defined as smokers.

c)：We are sorry for not expressing it clearly. Only education was not adjusted in mode 1, and education was adjusted in mode. 

d): Through correlation analysis, the result is obtained as follows. There was no significant correlation between BMI and lack of exercise (p-value of correlation significance = 0.057, correlation coefficient = 0.017). BMI and dyslipidemia were not significantly correlated (p-value of correlation significance= 0.306; correlation coefficient = 0.007). There is a significant correlation between lack of exercise and dyslipidemia (p-value of correlation significance=0.000). 

We are sorry that we did not reduce dimensionality to get the results. We performed a collinearity diagnosis and demonstrated that the variance inflation factor (VIF) of the family history of stroke, lack of exercise, dyslipidemia, atrial fibrillation, diabetes, BMI smoking, and other variables were 1.002, 1.009, 1.002, 1.005, 1.002, and 1.033, respectively, suggesting that there is no collinearity between these variables.

3.4 Interaction of risk factors: I suggest the authors to explain a bit more about Table 5, such as by comparing across PERI, AP and SI (not 'S').

Response:ThanksResponse: Thanks for your constructive comments. We have provided more explanations of the results in Table 5, including explanations of RERI, AP, and S.

4. Discussion and Conclusion

Overall, I appreciate the in-depth discussion of the main findings, but please shorten the discussion substantially. Besides, any conclusion about risk of stroke should be avoid given the cross-sectional design of the study. Therefore, I would prefer not to use 'risk factors' to denote the analyzed explanatory variables throughout the article either.

Response: Thanks for your constructive comments. We have shortened the length of the discussion and avoided using "Risk factor" to express our results in the article. 

Response: We are sorry for these grammatical errors. We have reviewed the professional language. To avoid the same mistake, we have double-checked the manuscript to ensure no such frequent mistakes in the revised version. 

Reviewer #2: This manuscript aimed to describe the prevalence of stroke in elderly population with hypertension in Shanghai and explore the risk factors of stroke. With a cross-sectional design, the authors employed multivariate logistic regression model to analyze the risk factors of stroke among hypertension population.

The results section, page 12 line 141 for those without stroke, 94.0% were married, while for those with stroke 91.7% were married. But the author said stroke patients had higher marriage rate.

Response: We are sorry for our carelessness. We have corrected this error. However, we no longer describe the prevalence of populations with different demographic characteristics due to suggestions from the editor and another reviewer. We still want to thank you for your valuable suggestions.

This manuscript is poor written. These are many typos and misuse of words. For instance,page9 line 50, “among which are independent, but also interact with each other”,page 10 line 87, what is ECG? Should give a full name when it occurred first time.page 11 line 111 “immediate family”?page 13 table 1 “secondary school?”

Response:I'mResponse: We are sorry for these grammatical errors and incorrect words. We have reviewed the professional language. To avoid the same mistake, we have double-checked the manuscript to ensure no such frequent mistakes in the revised version. According to your question, we have further explained these variables in detail. For example, the following content has been added to the “definition” section of the article:

ECG refers to Electrocardiogram;

The immediate family refers to grandfather, grandmother, parents, and siblings;

secondary school and above indicate having received a complete 9-year compulsory education or higher education in China.

The titles of all the tables need to be edited, it should clear describe what will be presented in the table.

Response: Thanks for your constructive comments. We have re-edited all table titles.

---

## [Decision Letter · Decision Letter 1]

2 Jul 2021

PONE-D-20-40953R1

Analysis of prevalence and influencing factors  of stroke in elderly hypertensive patients:

Based on the screening plan for the high-risk population of stroke in Jiading District, Shanghai

PLOS ONE

Dear Dr. Li,

Thank you for submitting your manuscript to PLOS ONE. After careful consideration, we feel that it has merit but does not fully meet PLOS ONE’s publication criteria as it currently stands. Therefore, we invite you to submit a revised version of the manuscript that addresses the points raised during the review process.

We look forward to receiving your revised manuscript.

Kind regards,

Y Zhan

Academic Editor

PLOS ONE

Journal Requirements:

Reviewers' comments:

Reviewer's Responses to Questions

**Comments to the Author**

1. If the authors have adequately addressed your comments raised in a previous round of review and you feel that this manuscript is now acceptable for publication, you may indicate that here to bypass the “Comments to the Author” section, enter your conflict of interest statement in the “Confidential to Editor” section, and submit your "Accept" recommendation.

Reviewer #1: (No Response)

Reviewer #2: All comments have been addressed

2. Is the manuscript technically sound, and do the data support the conclusions?

Reviewer #1: Partly

Reviewer #2: Yes

3. Has the statistical analysis been performed appropriately and rigorously? 

Reviewer #1: Yes

Reviewer #2: Yes

4. Have the authors made all data underlying the findings in their manuscript fully available?

Reviewer #1: No

Reviewer #2: Yes

5. Is the manuscript presented in an intelligible fashion and written in standard English?

Reviewer #1: No

Reviewer #2: Yes

6. Review Comments to the Author

Reviewer #1: Despite the revision, the authors did not manage to clarify how the study population was established in either the methods section in the manuscript or the response letter. As a consequence, I am not sufficiently convinced about the results interpretation and conclusion.

The statistics and results are relatively fine, but technical errors are universal (i.e. there is absolutely no way to investigate "risk" of an outcome in a cross-sectional design etc.) and must be corrected carefully before publication. Besides, results presentation is still unfortunately of low quality (inconsistent use of digit etc.)

Reviewer #2: My concerns had been fully addressed. No further revisions are necessary except language editing (minor revision)

7. PLOS authors have the option to publish the peer review history of their article (what does this mean?). If published, this will include your full peer review and any attached files.

Reviewer #1: **Yes: **Xiaoying Kang

Reviewer #2: No

---

## [Author Response · Author response to Decision Letter 1]

9 Jul 2021

Reviewer #1: Despite the revision, the authors did not manage to clarify how the study population was established in either the methods section in the manuscript or the response letter. As a consequence, I am not sufficiently convinced about the results interpretation and conclusion.

The statistics and results are relatively fine, but technical errors are universal (i.e. there is absolutely no way to investigate "risk" of an outcome in a cross-sectional design etc.) and must be corrected carefully before publication. Besides, results presentation is still unfortunately of low quality (inconsistent use of digit etc.)

Response:We have previously explained in detail in our manuscripts and reply letters how our research population was obtained, and the details are as follows：Since 2016, a comprehensive screening of all chronic patients in the Chronic Disease Management Bank of Jiading District, Shanghai has been conducted, and the general survey was performed on a community basis. The plan is mainly based on the chronic disease registration management database of the district. It registers all diagnosed chronic disease patients in the district. According to the principle of "informed, consent, and voluntary", the health department and the community strengthen publicity and guidance and implement full coverage screening for people aged over 35, meeting the inclusion criteria, and with hypertension registered in the chronic disease management database. The inclusion criteria of survey objects include: 1) permanent residents whose annual residence time is ≥6 months, and are over 35 years old; 2) clearly diagnosed as hypertension patients; 3) no contraindications related to screening; 4) patients who can sign an informed consent form by themselves or accompanied by their guardians and participate in all screening projects voluntarily. As of 2019, 26174 hypertension patients were randomly selected as survey subjects. After excluding people who are unwilling to cooperate and do not meet the inclusion criteria, we finally screened 21,902 hypertension patients, with a screening response rate of 83.69%. For these screened populations, the potential influencing factors of stroke and the occurrence of stroke were recorded. In the end, a total of 18,724 people with hypertension 60 years and older were selected as our study population. At present, the screening project is still ongoing. According to statistics, there are 155,829 hypertensive patients in the chronic disease registration management database in the district. A lot of manpower, material, and financial resources are required to achieve a comprehensive screening. Therefore, valuable information could be obtained through the investigation and analysis of the first stage to guide the follow-up screening and crowd intervention.

In addition, I would like to thank you for your recognition of our statistics and results. In addition, as you said, our cross-sectional study has no way to get the risk of the outcome, so we use the effect index OR to describe our outcome.And we carefully checked the description of the results to avoid using RISK to describe the results. Of course, this is also one of the limitations of our research, so we explained this in the limitations section. 

Finally, we carefully revised the results presentation, such as inconsistent use of digit.

Reviewer #2: My concerns had been fully addressed. No further revisions are necessary except language editing (minor revision)

Response:Thank you for your recognition of our research. We have polished and carefully reviewed the language of the manuscript.

---

## [Editor Report · Decision Letter 2]

14 Jul 2021

Analysis of prevalence and influencing factors  of stroke in elderly hypertensive patients:

Based on the screening plan for the high-risk population of stroke in Jiading District, Shanghai

PONE-D-20-40953R2

Dear Dr. Li,

We’re pleased to inform you that your manuscript has been judged scientifically suitable for publication and will be formally accepted for publication once it meets all outstanding technical requirements.

Kind regards,

Y Zhan

Academic Editor

PLOS ONE
---

## [Editor Report · Acceptance letter]

30 Jul 2021

PONE-D-20-40953R2 

Analysis of prevalence and influencing factors of stroke in elderly hypertensive patients:Based on the screening plan for the high-risk population of stroke in Jiading District, Shanghai 

Dear Dr. Li:

I'm pleased to inform you that your manuscript has been deemed suitable for publication in PLOS ONE. Congratulations! Your manuscript is now with our production department. 

Kind regards, 

on behalf of

Dr. Y Zhan 

Academic Editor

PLOS ONE